# A VARIANCE REDUCTION METHOD FOR NEURAL-BASED DIVERGENCE ESTIMATION

## ABSTRACT

A central problem in machine learning is the computation of similarity or closeness between two (data) distributions. The applications span from generative modelling via adversarial training, representation learning, and robustness in out-of-distribution settings, to name a few. A palette of divergences, mutual information, and integral probability metrics are indispensable tools for measuring the "distance" between distributions and these are made tractable in high dimensional settings through variational representation formulas. Indeed, such formulas transform an estimation problem into an optimization problem. Unfortunately, the approximation of expectations that are inherent in variational formulas by statistical averages can be problematic due to high statistical variance, e.g., exponential for the Kullback-Leibler divergence and certain estimators. In this paper, we propose a new variance penalty term that acts directly on the variance of each component of the statistical estimator. The power of the variance penalty is controlled by a penalty coefficient which trades off bias and variance. We tested the proposed approach on several variational formulas and synthetic examples and showed that the overall error is decreased about an order of magnitude relative to the baseline statistical estimator. Impressive results are obtained for Rényi divergence with large order values due to the improved stability of the proposed estimator. Furthermore, in real biological datasets we are able to detect very rare sub-populations with a moderate sample size. Finally, we obtain improved (in terms of objective measures) disentangled representation of speech signals into text, speaker, and style components via variance-penalized mutual information minimization.

## 1 INTRODUCTION

Divergences such as Kullback-Leibler (KL) divergence, $f$-divergences, Hellinger divergence, $\alpha$-divergences and Rényi divergences, which were initially developed in the fields of information theory and statistical physics, are indispensable tools in a growing number of machine learning applications. They have been used in adversarial training of generative models (Goodfellow et al., 2014; Nowozin et al., 2016), in the estimation of generalization errors (Esposito et al., 2021) and hypothesis testing (Broniatowski & Keziou, 2009), to name a few. Mutual information (MI), in particular, which is defined as the KL divergence between the joint distribution of a pair of variables and their marginals (and can be generalized to divergences other than KL), plays a crucial role in Bayesian networks and (conditional) independence (Cheng et al., 2002), self-supervised learning via contrastive losses (van den Oord et al., 2018; Le-Khac et al., 2020) as well as in representation learning (Hjelm et al., 2019; Chen et al., 2016).

Classical divergence estimators perform reasonably well for low dimensional cases, however they scale poorly to large, high dimensional datasets which are typically encountered in modern machine learning. The most compelling estimation approach of a divergence is via the optimization of a lower variational bound parametrized by neural networks. These lower bounds, which are likelihood-free approximations, are maximized in order to compute the divergence value at the optimizer. Well-known variational representations are the Legendre transformation of an $f$-divergence (Broniatowski & Keziou, 2006; Nguyen et al., 2010) as well as the Donsker-Varadhan (DV) variational formula (Donsker & Varadhan, 1983) for KL divergence and its extension to Rényi divergence (Birrell et al., 2020b). Their tractability stems from their objective functionals, which are computed from expected values and approximated using statistical averages from the available or generated samples.

Despite the scalability and tractability, the estimation of a divergence based on variational formulas is a notoriously difficult problem. One challenge stems from the potentially high bias, since any approximation for the worst case scenario requires an exponential number of samples in order to attain the true divergence value (McAllester & Stratos, 2020). Additionally, the statistical variance, which scales exponentially with respect to the divergence's value for certain variational estimators (Song & Ermon, 2019), is often prohibitively high. Focusing on the elevated MI, there are several further lower bounds (Barber & Agakov, 2003; Belghazi et al., 2018; van den Oord et al., 2018; Poole et al., 2019; Guo et al., 2021) and a few upper bounds (Cheng et al., 2020; Poole et al., 2019) which aim to provide more reliable estimates of MI in the low sample size regime. However, the majority of these MI estimators are not transferable to the general estimation of divergences and frequently produce instabilities during training which are further magnified by the small batch and/or sample size.

In this paper, we propose to reduce a divergence estimator's variance via an explicit variance penalty (VP) which is added to the objective functional. Our contributions are summarized as follows:

- We present a novel variance reduction penalty for $f$-divergence and expand it via the delta method to the nonlinear setting, including the DV formula for KL divergence as well as the variational formula for the Rényi divergences. The proposed VP is able to flexibly trade off bias and variance.

- We present numerical evidence on synthetic datasets that the proposed approach improves both mean squared error (MSE) and median absolute error (MedAE) in a range of sample sizes and types of divergences. Furthermore, we implemented the proposed VP in several other lower and upper bounds of MI, showing that our variance reduction approach is not restricted to particular variational formulas but it is generic and applicable to the majority of existing variational representations.

- When applied to real datasets, we demonstrate the ability of the proposed approach to reduce the variance of the estimated Rényi divergence, thus enabling the detection of rare biological sub-populations which are otherwise difficult to identify. Interestingly, the baseline estimator is unstable when the order value is above one, but it becomes stable when the VP is added.

- We also applied the VP to the disentangled representation learning of speech into its text, speaker, and style components. Results on objective evaluation metrics showed that the addition of the VP generally improves the training performance, as much as 18% relative to the baseline systems.

## 1.1 RELATED WORK

There are several general-purpose variance reduction techniques in Monte Carlo stochastic sampling, with the most popular approaches being antithetic sampling or more broadly coupling methods, control of variates and importance sampling (Robert & Casella, 2005; Glasserman, 2004; Srinivasan, 2013). These methods have not been explicitly applied for the variational divergence estimation problem. We speculate that either they are not applicable due to the unavailability of analytical probability density formulas or they are inefficient (e.g., the control of variates approach requires a second estimator and potentially a second parametric model in order to be applied).

Another way to reduce the variance is to restrict the function space to more smooth and/or controlled test (or critic) functions, balancing again between bias and variance. For instance, the restriction to Lipschitz continuous functions has the potential to reduce the variance since there exist favorable concentration inequality results for the Lipschitz space (Wainwright, 2019). In the GAN literature, Wasserstein GAN (Gulrajani et al., 2017) and spectral normalization (Miyato et al., 2018) impose Lipschitz continuity which resulted in signigicant gains in terms of training stability. Similarly, the restriction of test functions to an appropriately designed reproducing kernel Hilbert space could reduce the variance (Sreekar et al., 2020). Such approaches can be combined with our proposed variance penalties, as our formulation allows for general test-function spaces. However, we do not focus on this point here.

Given the importance of MI, several estimators aim towards improved statistical properties. Lower bounds such as MINE (Belghazi et al., 2018), which uses the DV variational formula with an expo-

nential moving average, NWJ estimator (Nguyen et al., 2010) and BA estimator (Barber & Agakov, 2003) as well as upper bounds such as CLUB (Cheng et al., 2020) still have high variance. InfoNCE (van den Oord et al., 2018) is one of the few MI estimators that has low variance, but at the cost of either high bias or high computational cost due to the need for many negative samples and thus large batch size. Poole et al. (2019) and Guo et al. (2021) aim to clarify the relationships and trade-offs between those variational bounds. A different approach to reducing variance is by appropriately working on the gradients of the objective function (Wen et al., 2020; 2021).

Finally, we discuss the approach of truncating the test function inside a bounded region as proposed in (Song & Ermon, 2019). The determination of the truncation threshold is quite difficult since it requires an a priori understanding of the log-likelihood ratio. Moreover, a high truncation threshold will not affect the estimation since a high threshold implies no real benefit in terms of variance reduction. On the other hand, a low threshold will result in large bias. Overall, using a high truncation threshold in order to avoid extreme values is a good practice even though it will have a limited impact on variance reduction.

## 2 BACKGROUND ON VARIATIONAL FORMULAS FOR RÉNYI AND $f$-DIVERGENCES.

While our variance reduction method can be applied to any divergence that possesses a variational formula, here our focus will be on the Rényi and $f$-divergences, including the KL divergence. For Rényi divergences an appropriate objective functional can be constructed from a difference of cumulant generating functions (Birrell et al., 2020b)

$$R_\alpha(Q\|P) = \sup_{g \in \mathcal{M}_b(\Omega)} \left\{ \frac{1}{\alpha-1} \log \mathbb{E}_Q[e^{(\alpha-1)g}] - \frac{1}{\alpha} \log \mathbb{E}_P\left[e^{\alpha g}\right] \right\}, \ \alpha \neq 0, 1. \tag{1}$$

Here $Q$ and $P$ are probability distributions on the set $\Omega$, $\mathbb{E}_Q$ and $\mathbb{E}_P$ denote the expectations with respect to $Q$ and $P$ respectively, and $\mathcal{M}_b(\Omega)$ is the space of bounded measurable real-valued functions on $\Omega$. For $f$ divergences, $f$ being a lower semicontinuous convex function with $f(1) = 0$, one has the well-known Legendre transform variational formula (Broniatowski & Keziou, 2006; Nguyen et al., 2010)

$$D_f(Q\|P) = \sup_{g \in \mathcal{M}_b(\Omega)} \left\{ \mathbb{E}_Q[g] - \mathbb{E}_P[f^*(g)] \right\}, \tag{2}$$

where $f^*(y) = \sup_{x \in \mathbb{R}} \{yx - f(x)\}$ is the Legendre transform of $f$. Here and in the following, the function of $g$ that is being optimized will be called the objective functional. Equation (2) can be generalized to the $(f, \Gamma)$-divergences (Birrell et al., 2020a), where $\Gamma \subset \mathcal{M}_b(\Omega)$ is a restricted test-function space

$$D_f^\Gamma(Q\|P) = \sup_{g \in \Gamma} \{ \mathbb{E}_Q[g] - \Lambda_f^P[g] \}, \tag{3}$$

$$\Lambda_f^P[g] = \inf_{\nu \in \mathbb{R}} \{ \nu + \mathbb{E}_P[f^*(g - \nu)] \}. \tag{4}$$

In particular, if $f_{\mathrm{KL}}(x) = x \log(x)$ corresponds to the KL divergence then

$$\Lambda_{f_{\mathrm{KL}}}^P[g] = \log(\mathbb{E}_P[\exp(g)]) \equiv \Lambda^P[g] \tag{5}$$

is the classical cumulant generating function and equation (3) (with $\Gamma = \mathcal{M}_b(\Omega)$) becomes the Donsker-Varadhan variational formula (Dupuis & Ellis., 1997, Appendix C.2)

$$D_{\mathrm{KL}}(Q\|P) = \sup_{g \in \mathcal{M}_b(\Omega)} \left\{ \mathbb{E}_Q[g] - \log \mathbb{E}_P[e^g] \right\}. \tag{6}$$

For general $f$, we will often write equation (3) as

$$D_f^\Gamma(Q\|P) = \sup_{g \in \Gamma, \nu \in \mathbb{R}} \{ \mathbb{E}_Q[g - \nu] - \mathbb{E}_P[f^*(g - \nu)] \} \tag{7}$$

and if $\Gamma$ is closed under the shifts $g \mapsto g - \nu$, $\nu \in \mathbb{R}$ then we can write it simply as

$$D_f^\Gamma(Q\|P) = \sup_{g \in \Gamma} \{ \mathbb{E}_Q[g] - \mathbb{E}_P[f^*(g)] \}. \tag{8}$$

In particular, if $\Gamma = \mathcal{M}_b(\Omega)$ then $D_f^\Gamma = D_f$. The generalizations of Rényi and KL divergence obtained by using a restricted space $\Gamma$ in place of $\mathcal{M}_b(\Omega)$ in equation (1) or equation (6) will be denoted by $R_\alpha^\Gamma$ and $D_{\mathrm{KL}}^\Gamma$, respectively.

## 3    STATISTICAL ESTIMATORS AND VARIANCE REDUCTION

Variational representations of divergences are especially useful for creating statistical estimators in a data-driven setting; a naive estimator is obtained by simply replacing expectations with the corresponding statistical averages in any of the equations (1), (2), (3), etc. More formally, the naive estimators can be written as $D_f^\Gamma(Q_n\|P_n)$, $R_\alpha^\Gamma(Q_n\|P_n)$, etc., where $\Gamma$ is some parameterized space of functions (e.g., a neural network), $Q_n$ and $P_n$ are the $n$-sample empirical measures from $Q$ and $P$ respectively (i.e., $\mathbb{E}_{P_n}[g] = \frac{1}{n}\sum_{j=1}^{n} g(X_j)$ where $X_j$ are i.i.d. samples from $P$ and similarly for $\mathbb{E}_{Q_n}$; we also assume that the samples from $Q$ and $P$ are independent of one another), and the divergences are expressed in terms of the variational formulas from Section 2. However, in practice these naive methods often suffer from high variance (Song & Ermon, 2019; Birrell et al., 2020b). We address this via variance-penalized divergences, which are constructed by introducing a variance penalty into the objective functional of the variational representation, e.g.,

$$D_f^\lambda(Q\|P) \equiv \sup_{g \in \mathcal{M}_b(\Omega)} \left\{ \mathbb{E}_Q[g] - \mathbb{E}_P[f^*(g)] - \lambda V[g; Q, P] \right\}, \tag{9}$$

where the variance penalty, $\lambda V$, is proportional to the variance of $\mathbb{E}_{Q_n}[g] - \mathbb{E}_{P_n}[f^*(g)]$ with strength $\lambda > 0$. Using this, we construct the following divergence estimator

$$\sup_{\eta} \left\{ \mathbb{E}_{Q_n}[g_\eta] - \mathbb{E}_{P_n}[f^*(g_\eta)] - \lambda V[g_\eta; Q_n, P_n] \right\}, \tag{10}$$

where $g_\eta$ is a neural network with parameters $\eta$. Similar variance penalties can be derived to other divergences with variational representations.

### 3.1    VARIANCE PENALTY

In this subsection we provide details on the variance penalty for $(f, \Gamma)$-divergences, the KL-divergence, and Rényi divergences. The same framework can be repeated to other divergences with a variational representation.

To introduce the variance penalty, first consider the $(f, \Gamma)$-divergence representation equation (8). Our goal is to penalize $g$'s for which $\mathbb{E}_{Q_n}[g]$ or $\mathbb{E}_{P_n}[f^*(g)]$ have large variance, hence we introduce a penalty term proportional to ($\mathbb{V}\mathrm{ar}_Q$ denotes variance with respect to $Q$, etc.)

$$\mathbb{V}\mathrm{ar}\left[\mathbb{E}_{Q_n}[g] + \mathbb{E}_{P_n}[f^*(g)]\right] = \frac{1}{n}\left(\mathbb{V}\mathrm{ar}_Q[g] + \mathbb{V}\mathrm{ar}_P[f^*(g)]\right). \tag{11}$$

Specifically, for $\lambda > 0$ we define the variance-penalized $(f, \Gamma)$-divergence

$$D_f^{\Gamma,\lambda}(Q\|P) \equiv \sup_{g \in \Gamma, \nu \in \mathbb{R}} \left\{ \mathbb{E}_Q[g - \nu] - \mathbb{E}_P[f^*(g - \nu)] - \lambda(\mathbb{V}\mathrm{ar}_Q[g - \nu] + \mathbb{V}\mathrm{ar}_P[f^*(g - \nu)]) \right\}. \tag{12}$$

As noted above, if $\Gamma$ is invariant under constant shifts then the optimization over $\nu$ can be omitted. A similar result to equation (12) can be derived for any objective functional that is a linear combination of expectations, e.g., integral probability metrics (Müller, 1997; Sriperumbudur et al., 2012) such as the Wasserstein metric.

For nonlinear objective functional terms of the generic form $G(\mathbb{E}_P[h(g)])$, such as appear in equation (1) and equation (6), we cannot compute the variance of the corresponding statistical estimator at finite $n$ but we can use the delta method to obtain the asymptotic variance

$$\lim_{n\to\infty} n\mathbb{V}\mathrm{ar}\left[G(\mathbb{E}_{P_n}[h(g)])\right] = (G'(\mathbb{E}_P[h(g)]))^2 \mathbb{V}\mathrm{ar}_P[h(g)]. \tag{13}$$

Thus, we propose for the nonlinear case to use the above asymptotic variance as a penalty and obtain the following variance-penalized KL and Rényi divergence variational formulas:

$$D_{\mathrm{KL}}^{\Gamma,\lambda}(Q\|P) \equiv \sup_{g \in \Gamma} \left\{ \mathbb{E}_Q[g] - \log\mathbb{E}_P[e^g] - \lambda\left(\mathbb{V}\mathrm{ar}_Q[g] + \mathbb{V}\mathrm{ar}_P[e^g]/(\mathbb{E}_P[e^g])^2\right) \right\}, \tag{14}$$

$$R_\alpha^{\Gamma,\lambda}(Q\|P) \equiv \sup_{g \in \Gamma} \left\{ \frac{1}{\alpha - 1}\log\mathbb{E}_Q[e^{(\alpha-1)g}] - \frac{1}{\alpha}\log\mathbb{E}_P[e^{\alpha g}] \right. \tag{15}$$

$$\left. - \lambda\left(\frac{1}{(\alpha-1)^2}\frac{\mathbb{V}\mathrm{ar}_Q[e^{(\alpha-1)g}]}{(\mathbb{E}_Q[e^{(\alpha-1)g}])^2} + \frac{1}{\alpha^2}\frac{\mathbb{V}\mathrm{ar}_P[e^{\alpha g}]}{(\mathbb{E}_P[e^{\alpha g}])^2}\right) \right\}.$$

**Remark 1.** *Both equation (11) and equation (13) suggest that the statistical estimators for the above penalized divergences should use a variance penalty strength that decays with the sample size $\lambda = \lambda_0/n$, though other forms of $n$-dependence may be useful in practice.*

Though the variance penalty introduces bias, as $\lambda \to 0$ the penalized divergence converges to the corresponding non-penalized divergence, as made precise by the following theorem.

**Theorem 2.** *Let $\Gamma \subset \mathcal{M}_b(\Omega)$. We have the following convergence results:*

$$\lim_{\lambda \to 0^+} D_{KL}^{\Gamma,\lambda}(Q\|P) = D_{KL}^{\Gamma}(Q\|P),\tag{16}$$

$$\lim_{\lambda \to 0^+} R_{\alpha}^{\Gamma,\lambda}(Q\|P) = R_{\alpha}^{\Gamma}(Q\|P),\tag{17}$$

*and if $f^*(y) < \infty$ for all $y \in \mathbb{R}$ then*

$$\lim_{\lambda \to 0^+} D_{f}^{\Gamma,\lambda}(Q\|P) = D_{f}^{\Gamma}(Q\|P).\tag{18}$$

*Moreover, under fairly general assumptions it holds that*

$$\lim_{\lambda \to \infty} D_{KL}^{\Gamma,\lambda}(Q\|P) = \lim_{\lambda \to \infty} R_{\alpha}^{\Gamma,\lambda}(Q\|P) = \lim_{\lambda \to \infty} D_{f}^{\Gamma,\lambda}(Q\|P) = 0.\tag{19}$$

**Remark 3.** *Note that the corresponding statistical estimators, $D_{f}^{\Gamma,\lambda}(Q_n\|P_n)$, etc., have additional bias due to the supremum over $g$. We present partial results on bias bounds in Appendix D.*

The proof of Theorem 2 is given in Appendix B for the zero limit and Theorem 9 for the infinity limit. The same proof techniques can be applied to other divergences with a variational characterization.

Finally, for non-zero $\lambda$ the penalized divergences (12), (14), (15) retain the divergence property and are therefore appropriate for quantifying the "distance" between probability distributions:

**Theorem 4.** *Under fairly general assumptions on $f$ and $\Gamma$ (see Appendix B for details) and letting $D^{\Gamma,\lambda}$ denote any of $D_{f}^{\Gamma,\lambda}$, $D_{KL}^{\Gamma,\lambda}$, or $R_{\alpha}^{\Gamma,\lambda}$ we have $D^{\Gamma,\lambda}(Q\|P) \geq 0$ and $D^{\Gamma,\lambda}(Q\|P) = 0$ if and only if $Q = P$.*

The proof of Theorem 4 can be found in Appendix B.

## 3.2 VARIANCE-REDUCED DIVERGENCE ESTIMATION ALGORITHM

We now propose the following divergence neural estimation (DNE) methods with variance penalty, generalizing equations (9)-(10).

**(DNE-VP$_\lambda$)** $$\sup_{\eta}\{H[g_\eta; Q_n, P_n] - \lambda V[g_\eta; Q_n, P_n]\}.\tag{20}$$

We will compare the above method to the non-penalized estimator (i.e., with $\lambda = 0$)

**(DNE)** $$\sup_{\eta} H[g_\eta; Q_n, P_n].\tag{21}$$

In the above, the test function space is a neural network $\Gamma = \{g_\eta, \eta \in E\}$ with parameters $\eta$ and $H$ denotes the objective functional of the divergence, e.g., for the Rényi divergences (1)

$$H_{\alpha}[g; Q, P] = \frac{1}{\alpha - 1}\log \mathbb{E}_Q[e^{(\alpha-1)g}] - \frac{1}{\alpha}\log \mathbb{E}_P\left[e^{\alpha g}\right], \quad \alpha \neq 0, 1\tag{22}$$

and for $f$ divergences (2)

$$H_f[g; Q, P] = \mathbb{E}_Q[g] - \mathbb{E}_P[f^*(g)].\tag{23}$$

Finally, $V$ is the variance penalty corresponding to the chosen divergence (see Section 3.1), e.g., for Rényi divergences

$$V_{\alpha}[g; Q_n, P_n] = \frac{1}{(\alpha-1)^2}\frac{\mathbb{V}\mathrm{ar}_{Q_n}[e^{(\alpha-1)g}]}{(\mathbb{E}_{Q_n}[e^{(\alpha-1)g}])^2} + \frac{1}{\alpha^2}\frac{\mathbb{V}\mathrm{ar}_{P_n}[e^{\alpha g}]}{(\mathbb{E}_{P_n}[e^{\alpha g}])^2}, \quad \alpha \neq 0, 1\tag{24}$$

and for $f$ divergences

$$V_f[g; Q_n, P_n] = \mathbb{V}\mathrm{ar}_{Q_n}[g] + \mathbb{V}\mathrm{ar}_{P_n}[f^*(g)].\tag{25}$$

We solve equation (20) via Adam algorithm (Kingma & Ba, 2014); a stochastic gradient descent method.

## 4 RESULTS ON SYNTHETIC DATASETS

Figure 1 presents the statistical estimation of Rényi divergence between two one-dimensional Gaussians which both have zero mean but different variance values. The order of Rényi divergence, $\alpha$, controls how much weight to put on the tails of the distributions, thus it can become very sensitive to the few samples from the tails. The same conclusion can be deduced from the variational formula (i.e., equation (1) where $\alpha$ multiplies the exponentials' argument). Therefore, a larger $\alpha$ value implies larger statistical variance. Indeed, high estimation variance is observed with DNE (upper leftmost panel of Figure 1) despite the fact that we applied truncation as proposed by Song & Ermon (2019) with truncation threshold set to 1. In contrast, the DNE-VP$_\lambda$ estimator with $\lambda = 0.1$ greatly reduces the statistical variance even when $\alpha$ is large (lower leftmost panel). For fairness, we imposed the same truncation operation in the output of DNE-VP$_\lambda$. We report a 80% reduction of variance for $\alpha = 2$ which becomes 99% for $\alpha = 10$.

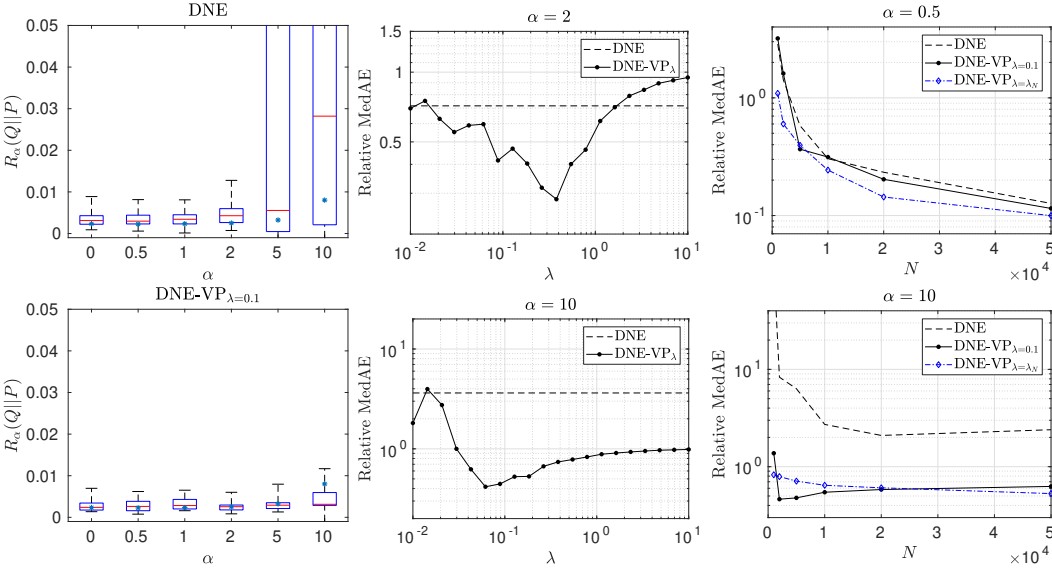

Figure 1: Comparison between the estimator without VP (DNE) and with VP (DNE-VP$_\lambda$) for Rényi divergence between two one-dimensional Gaussians with $Q = \mathcal{N}(0, 1.1)$ and $P = \mathcal{N}(0, 1)$. We use $N = 5K$ sample size, 512 as batch size and results are averaged over 100 i.i.d. runs. **Left column:** DNE and DNE-VP$_\lambda$ estimators for increasing values of $\alpha$. The variance of DNE becomes uncontrollably high for $\alpha > 3$. **Middle column:** Relative MedAE (the lower, the better) for varying penalty coefficient $\lambda$ and two values of $\alpha$. The relative MedAE for large values of $\lambda$ is close to one which implies that the estimated value of DNE-VP$_\lambda$ approaches zero. **Right column:** Relative MedAE for increasing sample size $N$. We additionally present a penalty coefficient that varies with sample size, shown in blue ($\lambda_N = \frac{500}{N}$ and $\lambda_N = \frac{2000}{N}$ for $\alpha = 0.5$ and $\alpha = 10$, respectively).

The proposed approach introduces an additional hyper-parameter, $\lambda$, which controls the strength of the VP. Our theory suggests that $\lambda$ should depend on the sample size (and perhaps also on the other parameters), therefore we perform two sets of experiments. In the first experiment, we explore the range of optimal values for $\lambda$ in terms of MedAE[1]. As is evident from the middle panels of Figure 1, $\lambda$-values in the vicinity of 0.1 are a reasonable compromise between variance and bias. In the second experiment, we demonstrate the performance in terms of MedAE as a function of the sample size, $N$. As suggested in Remark 1, monotone performance is obtained when $\lambda$ is inversely proportional to $N$ (blue dashed line in rightmost upper panel of Figure 1).

Our second synthetic example constitutes the estimation of MI using various approaches with and without VP. Here, we let $Q$ be a zero-mean multivariate correlated Gaussian random vector of dimension $d$. We impose element-wise correlation, i.e., $corr(x_i, x_{\frac{d}{2}+j}) = \delta_{i,j}\rho$ to the samples $x \sim Q$

---

[1]Recall that MedAE stands for median absolute error and it is a more robust-to-outliers metric.

where $i, j = 1, \ldots, \frac{d}{2}$ and $\delta_{i,j}$ is Kronecker's delta. With $P$ we denote the product of the marginals, which in this case is simply a zero-mean standardized multivariate Gaussian. Figure 2 presents the estimated MI per training step. We consider the Renyi-based MI with $\alpha = 0.5$ as well as the standard MI using the DV variational formula. Notice that these two variants result in different true values (black lines in Figure 2). The plotted results demonstrate the successful reduction of variance when VP is added to the objective functional. Interestingly, the extension of VP to InfoNCE and CLUB estimators (second row of panels in Figure 2) implies that our approach can be applied to any MI estimators, thus offering a general variance reduction framework. Bias, variance and MSE plots as well as several more experiments can be found in Appendix F.

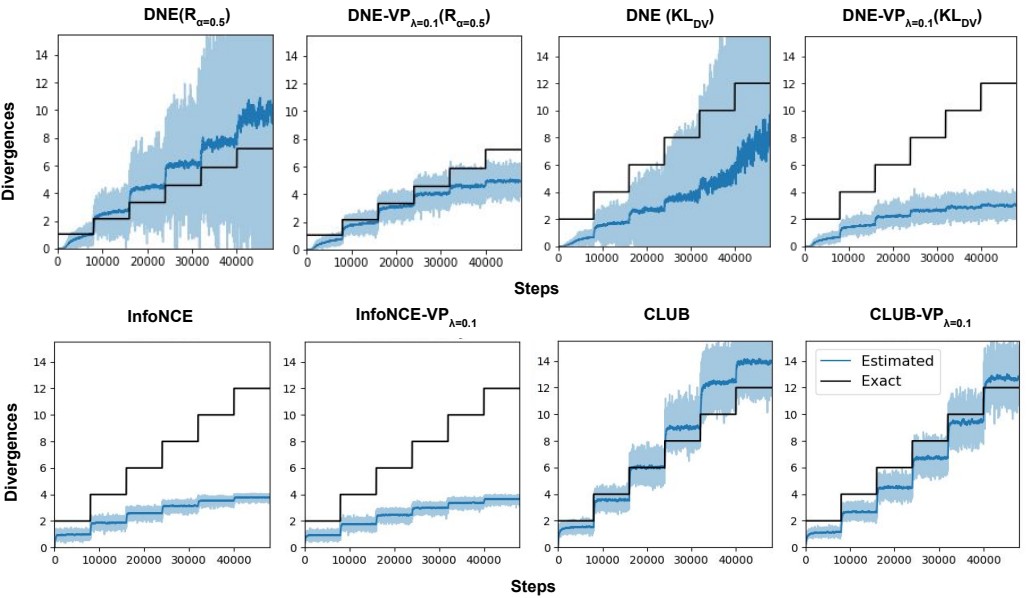

Figure 2: Performance comparison of several MI estimation approaches on a 40-dimensional correlated Gaussian random vector. The number of samples is set to $512K$ and batch size to 64. Panels with $R_{\alpha=0.5}$ in their titles present the Rényi-based MI with $\alpha = 0.5$ whereas the rest of the methods estimate the standard MI (i.e., the KL divergence). In each panel, the true values are shown as a step function (black line). The correlation coefficient of the Gaussian, $\rho$, for each step is: 0.3084, 0.4257, 0.5091, 0.5741, 0.6273 and 0.6717. The running estimates per minibatch are displayed as shadow blue curves. The dark blue curves shows the moving average of the estimated MI, with a bandwidth equal to 200 steps.

## 5 REAL DATA APPLICATIONS

### 5.1 DETECTING RARE BIOLOGICAL SUB-POPULATIONS

Using the dataset from Levine et al. (2015), we test the efficacy of DNE-VP$_\lambda$ in discriminating cell populations which are contaminated with a rare sub-population with distinguishable statistical properties. Specifically, we consider single-cell mass cytometry measurements on 16 bone marrow protein markers[2] (i.e., $d = 16$) coming from healthy and diseased individuals with acute myeloid leukemia. For each run we created three subsets of healthy samples with sample size $N = 20K$ which we denote by $P$ and one dataset as a mixture of 99% healthy and 1% diseased samples which is denoted by $Q$. Notice that the actual number of diseased samples is only 200 thus it is considered as a rare sub-population.

For Rényi divergence with $\alpha = 0.5$ (left panels in Figure 3), both DNE and DNE-VP$_\lambda$ are stable. Despite the improvement in the separation of the two histograms, the observed variance reduction

---

[2]Data was accessed from `https://community.cytobank.org/cytobank/experiments/46098/illustrations/121588`

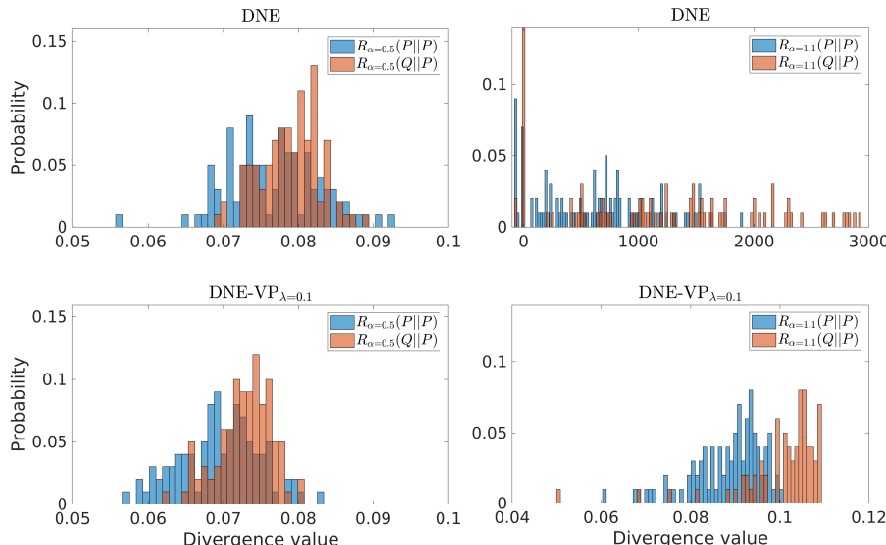

Figure 3: Comparison of DNE and DNE-VP$_\lambda$ estimators for Rényi divergence on biological data. The histograms of the estimated divergence value are constructed from 100 i.i.d. runs between datasets of $N = 20K$ samples each. Healthy dataset's distribution is denoted by $P$ whereas healthy + 1% diseased dataset's by $Q$. **Left column:** Rényi divergence with $\alpha = 0.5$. Neither DNE nor DNE-VP$_\lambda$ are able to discriminate between the healthy and the 1% contaminated dataset. **Right column:** Rényi divergence with $\alpha = 1.1$. For this $\alpha$ value, VP is compulsory for a stable estimation of Rényi divergence. Furthermore, we are able to discriminate between healthy and 1% contaminated distributions with high accuracy (87.5%).

of DNE-VP$_\lambda$ is minimal and not enough to discriminate between the healthy and the contaminated with 1% diseased samples distributions. When considering Rényi divergence with $\alpha = 1.1$, we observe that DNE fails to produce stable estimates. In contrast, DNE-VP$_\lambda$ always computes stable estimates. Additionally, the two histograms are satisfactorily separated, implying that larger values of $\alpha$ are crucial, provided there is a way to handle the statistical variance. For completeness, Table 1 reports the first and second order statistics of the histograms shown in Figure 3.

Table 1: Mean values and standard deviation for the histograms shown in Figure 3.

| Divergence | DNE | | DNE-VP$_{\lambda=0.1}$ | |
|---|---|---|---|---|
| | mean | std | mean | std |
| $R_{\alpha=0.5}(P\|P)$ | 0.0765 | 0.0066 | 0.0695 | 0.0053 |
| $R_{\alpha=0.5}(Q\|P)$ | 0.0789 | 0.0039 | 0.0720 | 0.0036 |
| $R_{\alpha=1.1}(P\|P)$ | 676 | 515 | 0.0890 | 0.0089 |
| $R_{\alpha=1.1}(Q\|P)$ | 1445 | 1165 | 0.1000 | 0.0120 |

## 5.2 Disentangled Representation Learning in Speech Synthesis

An important application of MI is disentangled representation learning. In the context of representation disentanglement, the extraction of meaningful latent features for high-dimensional data is challenging, especially when explicit knowledge needs to be distilled into interpretable representations. One popular approach to enforce representation disentanglement is via MI minimization. Moreover, a superior disentanglement will allow a greater degree of interpretability and controllability, especially for generative models maintaining high production capacity. In this section, we employ the proposed DNE-VP$_\lambda$ estimator for MI estimation in order to learn disentangled representation, and, particularly, in the context of speech synthesis and analysis.

A universal text-to-speech synthesizer can generate speech from text with speaker factor and speaking style similar to a reference signal. Previous works aimed to encode the information from reference speech into a fixed-length style and speaker embedding using trainable encoders (Wang et al., 2018; Tjandra et al., 2020; Chien et al., 2021; Tan et al., 2021). The major challenges for such speech synthesizers are controllability and generalisability, especially when trying to generalize the models with multiple speakers and multiple styles. During training, content information is leaked into the style embeddings ("content leakage") and speaker information into style embeddings ("style leakage"). Thus at inference, when the reference speech has different content from the input text, the decoder expects the content from the style vector ignoring some part of the content text. Moreover, speaker information could be expected from the style encoder leading to completely different speaker attribute. To alleviate that, Paul et al. (2021) suggested replacing the KL-based MI with Rényi-based MI and minimizing the Rényi divergence between the joint distribution and the product of marginals for the content-style and style-speaker pairs. However, reliable estimation of Rényi divergence was problematic due to high statistical variance. Taking advantage of the proposed variance reduction technique, we employ a VP term in the loss function which is denoted as DNE-VP$_\lambda$ ($R_\alpha$). By doing so, content, style, and speaker spaces become representative and (ideally) independent of each other. We introduce two variations of this framework: sum of three Rényi divergences DNE($R_{\alpha=0} + R_{\alpha=0.5} + R_{\alpha=1}$) (i.e., sum of the corresponding objective functionals) and DNE($R_{\alpha=0.5}$). We tested several different $\lambda$ values, aiming to reduce the statistical variance of the adversarial component. Notice that larger $\lambda$ values were helpful in this application.

Table 2: Objective evaluation tests. Lower scores indicate better performance.

| Methods | No Shuffle | | | Shuffle | | |
|---|---|---|---|---|---|---|
| | RMSE-F0 | MCD | WER(%) | RMSE-F0 | MCD | WER(%) |
| DNE ($R_{\alpha=0} + R_{\alpha=0.5} + R_{\alpha=1}$) | 28.59 | 5.35 | 21.6 | 45.75 | 6.39 | 28.7 |
| DNE ($R_{\alpha=0.5}$) | 28.59 | 5.27 | 18.3 | 47.26 | 6.60 | 26.6 |
| DNE-VP$_{\lambda=5}$ ($R_{\alpha=0} + R_{\alpha=0.5} + R_{\alpha=1}$) | 30.29 | **5.23** | 21.2 | 48.15 | **6.39** | 27.3 |
| DNE-VP$_{\lambda=10}$ ($R_{\alpha=0} + R_{\alpha=0.5} + R_{\alpha=1}$) | **27.76** | 5.36 | **18.1** | 47.62 | 6.48 | 28.7 |
| DNE-VP$_{\lambda=5}$ ($R_{\alpha=0.5}$) | 28.69 | 5.87 | **17.3** | 46.53 | 6.72 | **25.4** |
| DNE-VP$_{\lambda=10}$ ($R_{\alpha=0.5}$) | 29.71 | 5.33 | 22.8 | **45.47** | 6.54 | 26.2 |

We evaluate the performance of disentanglement strategies using three performance scores from 100 random samples shown in Table 2. Mel-cepstral distortion (MCD) measures the spectral distance between the synthesized and reference mel-spectrum features. Root mean squared error (RMSE) evaluates the similarity in F0 modeling between reference and synthesized speech. Lastly, the content preservation criterion is evaluated by word error rate (WER). During inference, we evaluate the performance on two conditions: 'no shuffle' and 'shuffle'. During inference, 'no shuffle' feeds the same reference speech into style and speaker encoders and its corresponding text to predict the speech features, whereas 'shuffle' feeds random speech. We observe that the proposed DNE-VP$_\lambda$ variants outperform baseline approaches without VP in terms of all evaluation metrics. Our proposed systems greatly reduced content leakage by improving the word error rate by approximately 5-18% relative to the baseline systems. Furthermore, RMSE-F0 and MCD scores show that the disentanglement module during training assists the TTS to achieve more accurate rendering of prosodic patterns as well as synthesizing proper speech content to its corresponding text without any significant leakage issues.

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
