# OpenReview forum: "A Variance Reduction Method for Neural-based Divergence Estimation"
_ICLR.cc/2022/Conference — ICLR 2022 Submitted_

### Official Review · Reviewer_KwzF · 2021-11-02

**Correctness:** 4
**Technical Novelty And Significance:** 3
**Empirical Novelty And Significance:** 3
**Recommendation:** 8
**Confidence:** 3

**Main Review:**

**Strengths:**

The paper clearly motivates the problem of high-variance in the estimation of the statistical divergences . Despite being heavy on notation, the manuscript is well-written and easy to follow, and the proposed solution is clearly laid-out.

The description of the method seemed mostly correct to me, although I did not verify the mathematics in detail.  The theoretical results are also a useful addition.

The experimental section assesses some of the properties of the proposed estimator sufficiently.


**Weaknesses:**

One of my concerns is that this paper does not compare to other variance reduction methods. The paper mentions that there are some other methods in the literature that apply techniques such as antithetic sampling to reduce the variance, but I did not see a direct comparison to such methods. How would they fair against the variance penalty method? Can the VP be used to supplement them?

Regarding the bias issue of the proposed estimator, would decaying the regularisation coefficient during the optimisation process help with alleviating this issue? If yes, I think this point can be explored in the experiments as a “best-of-both-worlds” approach.

I spotted some linguistics errors, e.g. "such as appear in equation (1)
and equation (8)” and  “there we also prove that”.

Minor point: Missing error bars on MedAE in Figure (1).


**Summary Of The Paper:**

This paper takes the variational view of the statistical estimation of statistical divergences. Standard variational formulas suffer from high variance when estimated with finite samples, this paper proposes to address the problem by adding a variance penalty regularisation term to the objective functional, thus trading variance for bias. The bias-variance trade-off is controlled via the regularisation coefficient. From this new formulation, the paper derives a new low-variance Neural Divergence Estimation Algorithm with the variance regularisation technique.

**Summary Of The Review:**

In my opinion, this work is original and well-presented. The paper discussed the presented method in-depth and the experimental evaluation was well-constructed. I think, overall, this work is a useful and significant contribution.

---

> ### Author Response · Authors · 2021-11-22
> **We address each point of the reviewer**
>
> 1) As already answered to a previous reviewer, the standard variance reduction methods are not applicable to our setup (eg, there is no obvious way to create antithetic samples from the real high-dimensional samples).
>
> 2) This is an interesting idea but it cannot be applied in a straightforward way. The reason is that an estimate for the bias is required but such estimate is unavailable since it requires the knowledge of the quantity we want to estimate.
>
> 3) We carefully reread the paper and correct all spotted linguistic errors.
>
> 4) Since we present the median values, we created a new figure in the supplementary with the 25\% and 75\% quartiles.

---

> > ### Comment · Reviewer_KwzF · 2021-11-28
> > **Acknowledgement of authors' response**
> >
> > I have read the rebuttal.

---

### Official Review · Reviewer_gwrQ · 2021-11-03

**Correctness:** 1
**Technical Novelty And Significance:** 1
**Empirical Novelty And Significance:** 2
**Recommendation:** 8
**Confidence:** 3

**Main Review:**

In this paper a highly topical problem is tackled, how to obtain low-variance divergence estimator. Authors decide to use variance penalty. Idea is quite good and synthetic experiments do seem to validate that reduction works.

I have only following points:
- About Fig 2, can you comment about why for CLUB VP does not seem to help so much as for DNE? Variance reduction, at least visually, seems to be quite minimal.
- It appears to be that there were no other variance reduction techniques compared, why not to compare against for example control variates?
- Why there is no Conclusions ?


**Summary Of The Paper:**

In this paper authors propose to an new neural-based divergence estimator using variance penalty.

**Summary Of The Review:**

Good contribution on an important topic.

---

> ### Author Response · Authors · 2021-11-22
> **We address each point of the reviewer**
>
> 1) CLUB is already a low variance estimator and the particular value for $\lambda_{VP}$ seems to make no difference. In principle, variance will be decreased if a larger value for $\lambda_{VP}$ is used (but with the cost of increased bias).
>
> 2) VP can be essentially considered as an addition to any variational bound and/or estimation method. The standard variance reduction methods from Monte Carlo literature (eg, control of variates or importance sampling) are not directly applicable to our setup.
>
> 3) Due to space limits we did not include a Conclusions section.

---

> > ### Comment · Reviewer_gwrQ · 2021-11-29
> > **Authors answered all my concerns.**
> >
> > Authors answered all my concerns sufficiently. However, addition of Conclusions would still be a good idea.

---

### Official Review · Reviewer_3SLV · 2021-11-06

**Correctness:** 2
**Technical Novelty And Significance:** 2
**Empirical Novelty And Significance:** 2
**Recommendation:** 3
**Confidence:** 4

**Main Review:**

(1) In Section 3.1, $G$ in Eq. (13) seems to override $f^*$ in previous equations, and $h$ is defined in the context. It is vague to understand how to derive Eq. (14) and Eq. (15) from Eq. (13). The authors should carefully take care of notations. For example, from (13) to (14), the author should at least denote $G=f^*$ and $h=\exp$.
(2) Theorem 1 is missing. Theorem 2 could be inappropriate and insufficient to be considered as a theorem.
(3) The assumption for Theorem 4. is not well described. The authors defer the details in Appendix B., but the proof which the author indicated only proves the divergence property of Renyi divergences instead of other divergences.
(4) In Section 3.2, $E$ is not defined and is ambiguous with $E$ which represents the expectation. The author should consider using other notations.
(5) Algorithm 1 is overly simplified and incorrect. Line 3 in Algorithm 1 implies that the proposed variance penalty is not included during optimization; therefore the variances would not be suppressed. Also, since $\alpha$ is used by Renyi divergence, the authors should denote the parameter of Adam by another symbol other than $\alpha$.
(6) Figure 1 did not comprehensively analyze the trade-off of bias and variance. In addition, it is necessary to explain why the authors choose MedAE instead of MSE.
(7) The trade-off between bias and variance is unsatisfactory. For instance, VP does not affect InfoNCE. Moreover, to greatly reduce variance, DNE-VP exhibits large biases.
(8) The second synthetic experiment seems problematic. First, as we know that InfoNCE is bounded above by $\log N$, where $N$ is 64 according to the caption in Figure 2. Therefore, InfoNCE should saturate to about 4.159, but the authors' experiment shows that the estimated MI saturates below 4. Second, the author examined estimators under different scales of the ground-truths.
(9) The authors might have overlooked that as they adopted the Delta method, the estimators should be guaranteed to be ${\it consistent}$. The authors did not prove that all estimators included in their experiments are consistent.
(10) The authors did not introduce the model architectures and datasets used in the speech synthesis experiments (for both speech synthesizer and automatic speech recognition systems).  Without such information, readers cannot reproduce the results easily, and thus the results reported in the paper are not convincing.


**Summary Of The Paper:**

The paper proposes to adopt a variance penalty term to manipulate the divergence estimators. Experiments first investigated the correlations of the variance penalty and the bias and variance trade-offs. Then, the authors tested the proposed approaches on two real-world application (biological and speech synthesis) datasets. The results show that the proposed approach can effectively reduce the variance as compared to systems without including the variance penalty.

**Summary Of The Review:**

The proposed idea is reasonable. However, some key information (including notation definitions and experimental setups) is not well described, which may make readers difficult to follow the main ideas and reproduce the results.

---

> ### Author Response · Authors · 2021-11-22
> **We address each point of the reviewer**
>
> 1) Actually, $h$ engulfs both $f^*$ in $f$-divergence and $\exp$ in DV formula. Similarly, $G$ can be either $\log$, $id$ or any other function outside the expectation. Eq. (13) is a generic formula that can sufficiently describe the elements that constitute almost any variational bound. We added a sentence to make (13) clear.
>
> 2) It's a numbering issue of ICLR format. Moreover, we beef up Theorem 2 by gathering both extremes cases (ie, both $\lambda\to 0$ and $\lambda\to\infty$) into it.
>
> 3) We did present them in the supplementary. The proofs corresponded to the proofs of Theorems 7 and 8 in the supplementary. Concerning the required assumptions, they are fairly general and standard in order to prove the divergence property.
>
> 4) $E$ always denotes the expectation. In order to increase the clarity we used $\mathbb E$ for the expectation in the updated manuscript.
>
> 5) We removed the pseudo-code since it is somewhat trivial and we will provide the actual python code.
>
> 6) MedAE is a more robust statistic that requires less repetitions for reliable estimates. For the sake of completeness, we added a figure in the supplementary (Fig. 4) which presents the variance, the (squared) bias and the MSE.
>
> 7) The primal purpose of this example (and the respective Fig. 2) is to demonstrate that the proposed VP framework is easily transferable to other variational bounds. Therefore, we did not search for the optimum hyper-parameter value and use a typical value for $\lambda=0.1$. As in the other presented examples, there is a large range of $\lambda$-values where the proposed approach produces improved performance (eg, in terms of MSE).
>
> 8) We used the code from \url{https://github.com/Linear95/CLUB} which accompanied the ICML2020 paper "CLUB: A Contrastive Log-ratio Upper Bound of Mutual Information". The argument by the reviewer is only partially correct since the upper bound $\log N$ is reached only if the parametric model has infinite capacity. This is never the case in practice because the test functions are approximated by parametric models (eg, neural networks) therefore any estimator will typically compute a value below the upper bound.
> Concerning the second point on the ground truth values of the divergences, we chose to compute the divergences for the correlation coefficient levels. Moreover, 3 out of 4 cases in Fig. 2 corresponded to same KL divergence. It is only the Renyi divergence with order $\alpha=0.5$ that took different values.
>
> 9) The proposed estimators are consistent but indeed we didn't write explicitly these asymptotic limits. Actually, the standard tools used in previous studies are adequate to prove asymptotic consistency. However, as it has been shown before, the variance due to finite sampling is one of the most crucial factors that determine the performance of a divergence estimator. Finally, we would like to point out that the Delta method is not used as an approximation of our estimator. Rather it is used to inform which is the proper variance penalty term. Therefore, it can't affect any consistency outcome.
>
> 10) It is the exactly the same as in the referenced Interspeech paper. In particular, the speaker encoder training has been conducted on the datasets LibriSpeech, VoxCeleb1 and VoxCeleb2 which contain utterances from over 8 thousands speakers. TTS model is trained using VCTK English corpus and LJSpeech database is used for the “warm-start” approach. With respect to the architectures, we implemented TransformerTTS as our backbone speech synthesizer. As for the automatic speech recognize system, we used Google’s open-source automatic speech recognizer.
> The detailed architectural descriptions can be found in the cited Interspeech paper.

---

### Official Review · Reviewer_JtE9 · 2021-11-06

**Correctness:** 3
**Technical Novelty And Significance:** 2
**Empirical Novelty And Significance:** 2
**Recommendation:** 3
**Confidence:** 4

**Main Review:**

The paper present an effective method to reduce the variance of divergence estimators based on variational bounds. In summary, the strengths and weaknesses of the paper are listed below:

** strengths **
- The investigated problem is relevant and the paper is well-written and the contributions clearly presented.
- For the considered synthetic scenarios the proposed approach improves the mean square error which seems to validate the underlying hypothesis of the regularzation term introduced to correct the variance.
- Although the main idea of exploiting the asymptotic expression of the variance (based on the delta method) to introduce a penalty term that controls the variance is not surprising, the application within the framework of estimation of diverges based on variational bounds and the effectiveness of the method seems to be interesting.

** weaknesses **
- The main misconception of this work relies on the fact that the synthetic experiments provided in Section 4 are too simplistic to be able to validate the proposed method. More precisely,  (i) the regimes under which the estimators are investigated are rather optimistic since a large amount of samples are considered (e.g. 512K); (ii) there is no experiments on multimodal (mixtures) distributions or other distributions than Gaussian distributions (e.g., there are many possible examples where the underlying information measure does not change but the the distribution of the samples can be very much different by introducing complex transformations); (iii)  there is no studies analysing the effects of the batch size, the effects of the underlying optimisers used to obtain the variational  lower bounds (e.g., what about the use of batch normalisation?; (iv) there is no investigation of the varice reduction under realistic high dimensional datasets (e.g. SVHN, imagenet, etc.)
- The empirical results are compared with some previous works but not state-of-the-art estimators. For example, CLUB has been improved in [A] and see references in therein for other missing methods and possible additional non-Gaussian scenarios to study. Indeed, this estimator is also applied to the problem of disentanglement and thus, it is relevant to compare with the results obtained on Section 5.2.
A comparison with SOTA methods is fundamentally important to validate the effectiveness of the proposed method. Which optimiser is used for the experiments in Fig. 2 ?
- The resulting estimator after the variance regularization should also be compared to the work in [B] which  studies for a particular case the asymptotic variance the f-divergence estimators. What can be said about the variance of the proposed estimation with the framework of [B] ?
- It is claimed that "the majority of the existent estimators for MI are not transferable to the general estimation of the divergences and frequently produce instabilities during training". This claim is not valid or at least not clear enough since the estimation methods that will be consider in this paper follow the same ideas (i.e., they are based on variational bounds).
- Similar comments apply to the empirical results presented in sections 5.1 and 5.2. For example, there is no comparison of the variance reduction compared to the reference [C] which is particular relevant for the investigated framework in Section 5.2.

** References **
[A] A Novel Estimator of Mutual Information for Learning to Disentangle Textual Representations by P. Colombo et al, ACL 2020.
[B] Practical and Consistent Estimation of f-Divergences by Paul K. Rubenstein et al., NeurIPS 2019.
[C] Adversarial Disentanglement of Speaker Representation for Attribute-Driven Privacy Preservation by Paul-Gauthier Noé et al., Interspeech 2021


**Summary Of The Paper:**

This paper introduces a novel variance regularization to reduce the statistical variance of the  variational based estimators for f-divergences and Renyi's divergence. The proposed regularization is based on the well-known delta method which provides the asymptotic variance and  depends on the specific from of the variational bound under consideration. The approach is tested on different synthetic data sets for which the numerical results show that the variance is decreased relatively to the baseline estimator. In particular, this effects appears be more visible for high-orders of the Renyi's divergence. Finally, an application is provided for real biological datasets, disentanglement for speech signals into text, speaker and style components.

**Summary Of The Review:**

The paper introduces some novel and interesting ideas. However, the numerical experiments  provided  in this paper are neither complete nor  always relevant. In particular, comparison with state-of-the-art estimators are clearly missing, the synthetic datasets are too simplistic  to validate the proposed method, the regimes (in terms of samples) are not relevant and many other effects related to the choice and impact of the optimiser, batch size, etc... are missing. I cannot recommend the acceptance of this paper.

---

> ### Author Response · Authors · 2021-11-22
> **We address each point of the reviewer**
>
> 1.i) We did present experiments with less (or even more) sample sizes. We presented various sample sizes in the first synthetic example (right column of Fig. 1) and similarly for the second synthetic example (Figs. 4-10 in the Supplementary). Moreover, the sample size for both real data experiments is of order $O(10^4)$.
>
> 1.ii) The synthetic distributions were intentionally meant to be simple. Actually, high statistical variance is not just a matter of high dimensions or the complexity of the distributions since large divergence values result in huge variance independently from the type or shape of the distributions.
> Even for 1d Gaussian examples (eg, Renyi divergence with large order value), the value as well as the variance of a divergence  estimator could become infinity. Moreover, the real data experiments demonstrated far more complicated distributions since the biological dataset (Fig. 3) is multi-modal while speech signals are simultaneously non-linear, non-stationary and non-Markovian.
>
> 1.iii) Figs. 4-10 in the Supplementary presented and compared various batch and sample sizes.
>
> 1.iv) The raw speech signals are also very high-dimensional. An utterance with length of few seconds corresponds to a sample of dimension between 20K and 100K. Of course, disentanglement is enforced at the embedding space (256 dimensions for speaker, 256 dimensions for text and 256 dimensions for style) but the same holds for images, too (meaning that disentanglement is applied at the embedding space).
>
> 2) We incorporated our variance reduction penalty in NWJ, MINE, InfoNCE, CLUB, CLUBSample and L1OutUB estimators and compared with all of them. These diverse estimators are considered sate-of-the-art and have been used in previous studies for comparison purposes. The majority of the results are shown in the Supplementary (Figs. 4-10) due to space limitations. The interesting reference suggested by the reviewer is indeed a new estimator for MI based on an upper bound (as in CLUB) but there is not a single mention on its variance or related improvements in terms of variance reduction.
> Finally, we would like to point out that our main goal with Fig. 2 is to demonstrate the generality of the proposed VP by showing that it can be easily transferred and implemented to other MI estimators.
> Concerning the optimizer, we have used Adam in all of our experiments.
>
> 3) The suggested framework has interesting theoretical analysis and some of the asymptotic tools might be useful in general. However, they assume the explicit knowledge of the density for $P$ and partial knowledge of the density for $Q$. We do not consider such strong assumptions about the densities in our study. Nevertheless, we did present some standard asymptotic results (eqs. (11) and (13)).
>
> 4) What we meant is that the majority of the available estimators are focused on mutual information, ie, the KL divergence. Indeed, they have proposed variational bounds for KL divergence which are not transferable in a straightforward way to other divergences (eg, f-divergence or Renyi divergence). On the other hand, the proposed variance penalty can be easily calculated as a statistical quantity irrespective of the utilized variational bound.
>
> 5) Our method is not comparable with the suggested work for two main reasons: (a) The application in the referenced paper corresponds to attribute driven privacy preservation (especially sex attribute) in speaker voice representation which is different (they remove sex information from the embedding while we break out the information into speaker, text and style components). (b) Subsequently, their approach is substantially different (they use auxiliary classifiers for sex variable while we perform mutual information minimization). Moreover, our work focused on unsupervised learning whereas the authors of the referenced paper considered a semi-supervised learning approach (they know the gender for each utterance).

---

> > ### Comment · Reviewer_JtE9 · 2021-11-28
> > **Acknowledgement of authors' response and efforts in reviewing the paper.**
> >
> > I have carefully check both the replies made by the authors and the revised paper. However, I cannot agree with some the responses. For instance, point (5). Still, I think the propose method requires further and extensive numerical analysis before being  ready for publication.

---

### Decision · Program_Chairs · 2022-01-20

**Decision:**

Reject

**Comment:**

This paper proposes a new class of divergences that are also sensitive to the variance of the estimator. The proposed additional variance penalty term introduces a bias term and acts directly on each component of the statistical estimator. By choosing the penalty parameter one can trade bias versus variance.

The results on synthetic examples look promising and suggest that with this technique, it is feasible to decrease the estimation error relative to the baseline statistical estimator. This is demonstrated to be particularly pronounced for certain Renyi divergences in the large order parameter alpha regime. Two applications (detection of subpopulations and disentangled representation learning in speech) are provided.

The opinions about the work were fairly divided. Both positive reviews have lower confidence and are rather short and do not fully justify the high rating. Two high confidence reviews are negative and raise several critical points. In a nutshell, reviewer JtE9 complains mainly about the insufficient experimental evaluation while 3SLV raises several concerns regarding readability, mathematical notation, lacking details of the proofs, as well as technicalities regarding the consistency. The authors have partially answered the concerns.

While there seems to be a consensus that the paper is interesting and makes a valid contribution, the introduction of the VP term defines a new estimation problem and both the choice and interpretation of lambda becomes critical. In particular, the key question is understanding the effect of lambda for various tasks where divergence estimation is crucial and I am not fully convinced if the chosen applications are the best for convincingly demonstrating the utility as these require somewhat application specific motivation. I would rather see result of standard benchmark datasets (such as estimating the divergence between two subsets of MNIST images to detect subtle distribution shifts).

The synthetic experiments are good but this section could be improved as well to get the message accross. Rather than delving directly to the findings, this section could first justify what needs to be measured and what are the control variables (number of samples, Renyi order etc)

In light of the comments raised by the reviewers, I feel that this paper can benefit from a further iteration and clarification of the experimental section before being accepted to a venue like ICLR.